# Leveraging mHealth usage logs to inform health worker performance in a Resource-Limited setting: Case example of *mUzima* use for a chronic disease program in Western Kenya

**Simon Savai**[1]*, **Jemimah Kamano**[2,3], **Lawrence Misoi**[3], **Peter Wakholi**[4], **Md Kamrul Hasan**[5], **Martin C. Were**[6]*

1 Institute of Biomedical Informatics, Moi University, Eldoret, Kenya, 2 School of Medicine, Moi University, Eldoret, Kenya, 3 Moi Teaching and Referral Hospital, Eldoret, Kenya, 4 School of Computing and Informatics Technology, Makerere University, Kampala, Uganda, 5 Department of Computer Science, Vanderbilt University, Nashville, Tennessee, United States of America, 6 Department of Biomedical Informatics and Medicine, Vanderbilt University Medical Center, Nashville, Tennessee, United States of America

* mssavai@gmail.com (SS); martin.c.were@vumc.org (MCV)

**Data Availability Statement:** All data are contained in the manuscript and Supporting Information files.

## Abstract

### Background

Health systems in low- and middle-income countries (LMICs) can be strengthened when quality information on health worker performance is readily available. With increasing adoption of mobile health (mHealth) technologies in LMICs, there is an opportunity to improve work-performance and supportive supervision of workers. The objective of this study was to evaluate usefulness of mHealth usage logs (paradata) to inform health worker performance.

### Methodology

This study was conducted at a chronic disease program in Kenya. It involved 23 health providers serving 89 facilities and 24 community-based groups. Study participants, who already used an mHealth application (*mUzima*) during clinical care, were consented and equipped with an enhanced version of the application that captured usage logs. Three months of log data were used to determine work performance metrics, including: (a) number of patients seen; (b) days worked; (c) work hours; and (d) length of patient encounters.

### Principal findings

Pearson correlation coefficient for days worked per participant as derived from logs as well as from records in the Electronic Medical Record system showed a strong positive correlation between the two data sources (r(11) = .92, p < .0005), indicating *mUzima* logs could be relied upon for analyses. Over the study period, only 13 (56.3%) participants used *mUzima* in 2,497 clinical encounters. 563 (22.5%) of encounters were entered outside of regular

**Funding:** This work was made possible by the support of the American people through the United States Agency for International Development (USAID, grant number 7200AA18CA00019) and the Norwegian Agencies for Development Cooperation under the NORHED program (Norad: Project QZA-0484). The funders had no role in study design, data collection and analysis, decision to publish, or preparation of the manuscript.

**Competing interests:** I have read the journal's policy and the authors of this manuscript have the following competing interests. At the time of submission MW was a Section Editor for PLOS Digital Health.

work hours, with five health providers working on weekends. On average, 14.5 (range 1–53) patients were seen per day by providers.

## Conclusions / Significance

mHealth-derived usage logs can reliably inform work patterns and augment supervision mechanisms made particularly challenging during the COVID-19 pandemic. Derived metrics highlight variabilities in work performance between providers. Log data also highlight areas of suboptimal use, of the application, such as for retrospective data entry for an application meant for use during the patient encounter to best leverage built-in clinical decision support functionality.

### Author summary

Mobile health (mHealth) applications have gained significant penetration to support health care in low- and middle-income countries. Beyond improving care, these applications can help to strengthen the health system, but are currently not optimally employed for this goal. We explored whether we could leverage usage log data, which were captured as health providers were using a mobile application called *mUzima*, to inform health worker performance in Western Kenya. We were able to reliably demonstrate that the log data can provide detailed information of the days and hours worked, number of patients seen per day, and how the application was being used to inform many areas of improvement related to health worker performance. We also observed large differences in work performance between health providers, and work performed outside of official work hours and on weekends. Some of the health providers also used the application sub-optimally for entering patient data after the patient visit, as opposed to using the application during the patient encounter to best leverage built-in clinical decision support functionality. This study offers an approach to cost-effectively augment health worker supervision mechanisms that have been particularly challenging during the COVID-19 pandemic and for providers distributed over wide geographical areas.

## Introduction

Health service delivery in low- and middle-income countries (LMICs) has often been characterized by inadequate health-worker performance, resulting in poor quality and low uptake of health services [1]. Poor health-worker performance is reflected in under-achievement towards clinical targets, absenteeism, low motivation, poor service quality, and fabrication of health data [2–5]. Among reasons cited as contributors to this poor performance include: inadequate numbers of health personnel leading to work overload [6], poor working conditions [2]. and inadequacy of supervision [6].

While supportive supervision is considered an essential intervention for improving health-worker performance [1,7,8], such interventions are only effective with consistent and targeted interactions between workers and their supervisors. Unfortunately, in many LMIC settings, supportive supervision can be difficult to achieve. Oftentimes, the number of supervisors is inadequate to support all workers, with supervisors typically responsible for covering multiple facilities and wide geographical areas. The challenges around direct monitoring of work

performance and supportive supervision are further exacerbated with emergence of the COVID-19 pandemic that has limited travel and in-person contact between personnel. With restrictions and lockdowns imposed in many localities, it becomes difficult for supervisors to travel to communities to support workers, and vice versa. It is imperative that innovative mechanisms for effective performance monitoring and supportive supervision are explored. These approaches would require judgments to be made on a continuous basis, with availability of good quality and timely information at both individual health worker and at group levels.

mHealth technologies are now in broad use to support health workers in LMICs [9–12]. In many settings, workers and their supervisors use smartphone-based applications (apps) primarily for care coordination, data capture, retrieval of patient data, and decision support. [9,10,13] The apps can also support many other functions such as secure clinical messaging, tele-consultation and geo-location services. These mHealth solutions have an ability to collect paradata, defined as "process data documenting users' access, participation, and navigation through an mHealth application" [14–16]. Unfortunately, despite the rich quality of paradata that can be securely collected through usage logs from mHealth apps used by health workers, these data have not been leveraged to inform health work-performance and for supportive supervision. In fact, to date, the use of mHealth paradata have only been limited to evaluations of users' engagement with mHealth applications [14,17].

We hypothesized that mHealth applications, through collected paradata could be used to improve performance monitoring and supportive supervision. In this paper, we describe an evaluation that uses mHealth application-derived logs (paradata) from a demonstrative application, *mUzima* [18], to inform work patterns for health care workers in an LMIC setting who do not have frequent direct in-person contact with their supervisors. The employed approach has broad applicability across mHealth applications, extending the use of mHealth solutions to support health systems strengthening initiatives in LMICs.

## Materials and methods

### Study overview

This study involved several steps as outlined in **Fig 1**. In the first step, the widely deployed *mUzima* mobile application was enhanced to enable capture of usage logs during use of the application providers [18]. Health workers, who were already familiar with *mUzima*, were then recruited and consented to use the enhanced mUzima application. Usage logs were collected for a study period of three months. At the end of this period, data were extracted from collected usage logs and from the associated electronic medical record (EMR) system at the study sites. The logs were analysed to gain insight into health worker performance, including: (a) days and times worked by providers, (b) workday length, (c) clinical encounter length, and (d) number of patients seen. Details for the steps are provided below.

### Software development

*mUzima* is a robust open-source Android application for use by health providers to primarily capture data as part of patient care. It also has features for reviewing patient's historical data, as well as for computerized clinical decision support, among others (**Fig 2**) [18]. The application

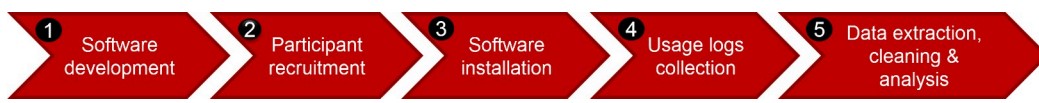

**Fig 1. Study overview.**

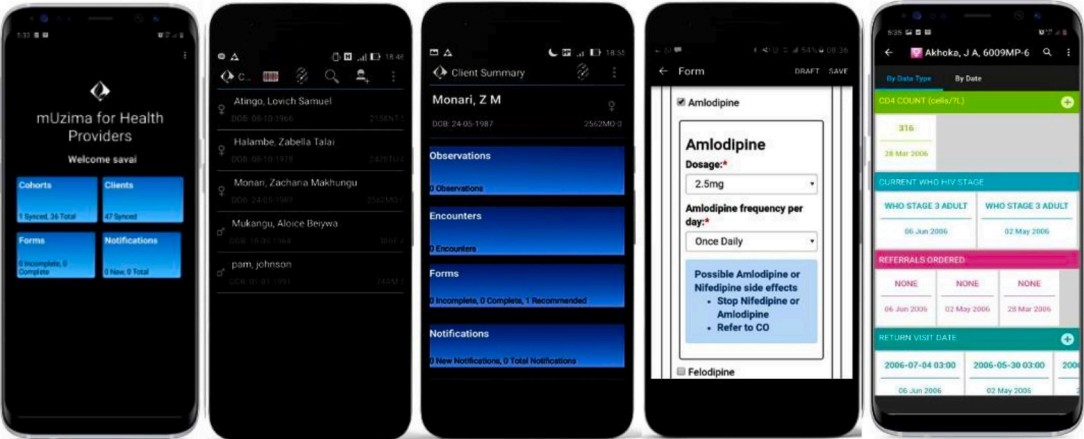

**Fig 2. Sample screenshot of the mUzima mobile application.**

works on Android-based smartphones and tablets. *mUzima* is configured to interoperate with the OpenMRS EMR system that is widely used in over 40 LMICs [19,20]. *mUzima* has offline capability and is thus used widely in communities and within facilities that do not have consistent online access to the EMR system.

For this study, the *mUzima* application was enhanced to generate, store, and transmit event logs as the application was in use. As the goal of logging was to capture activities conducted by users on the mobile application, an event log was generated whenever a user navigated to a user interface (UI) page on the mobile device. Each event log had a standard structure, with key elements including: (a) event/activity type, (b) a unique identifier of the UI page, (c) a unique identifier of the mobile device, (c) a unique identifier of the mobile app user, (d) GPS geolocation information, (e) timestamp of the device, (f) event timestamp, (g) transmission server timestamp, and (h) other context details such as machine generated patient identifier and recorded data unique identifier. A full list of the elements that constituted an event log, as well as description of each element can be found in **S1 Appendix**. **S2 Appendix** details all the 76 types of event logs that were captured as a user navigated through the application—examples of these event logs included: '*view_client_list*', '*save_complete_encounter_form*' and '*open_registration_form*'.

Recognizing that all users and programs would not want usage logs captured, a setting was added to the *mUzima* application to easily turn the logging feature on or off. An *Event Log Server* was configured to securely receive generated event logs that were transmitted from the *mUzima* application. Transmission of these logs used the same secure https-based mechanism already in place to transmit clinical data from *mUzima* to the EMR system. The *Event Log Server* was based on the *MongoDB* NoSQL database, which provided flexibility and scalability in storing large volumes of data (**Fig 3**) [21].

## Study setting

Academic Model Providing Access To Healthcare (AMPATH) was established in Kenya in 2001, and has developed an HIV care system in Western Kenya that serves over 100,000 patients, with a robust data system, the AMRS [22]. Initially started to support HIV care, the AMPATH program has expanded its clinical scope of work in several counties in western Kenya, to address comprehensive primary care, including non-communicable diseases in its chronic disease management programs. Under the Primary-health Integrated Care project for

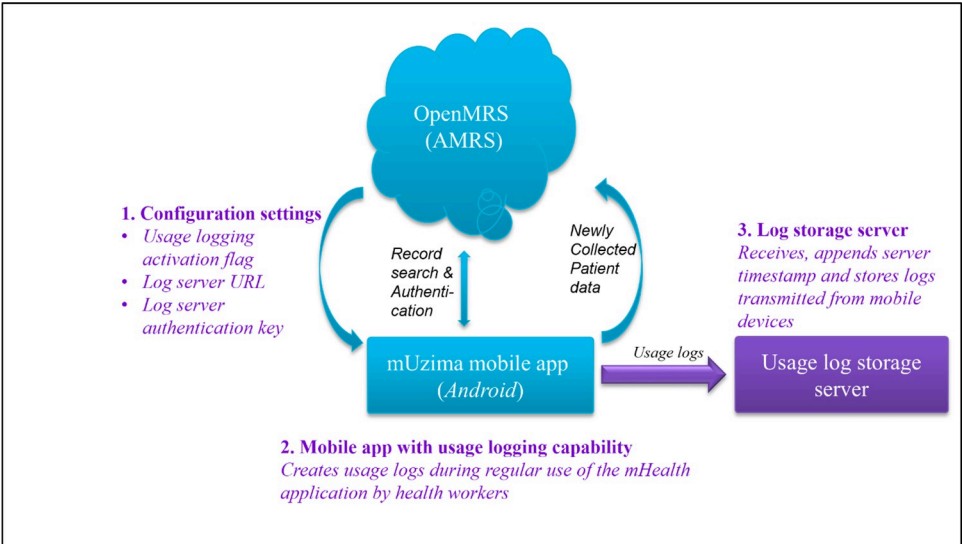

**Fig 3. Illustration of the interaction between components of the developed software system.**

Chronic diseases (PIC$_4$C), AMPATH uses task shifting of chronic diseases management to nurses and clinical officers. The goal of this task shifting is to improve access to chronic disease care services across geographically decentralized facilities [23]. This study was conducted in two AMPATH supported counties, *Busia* and *Trans Nzoia*, which were at the time supported by the PIC$_4$C program that sought to integrate hypertension and Diabetes care into the established primary healthcare platforms while strengthening referral systems. Within the study setting official clinician work hours were between 8 am to 5 pm from Monday to Friday.

## Study participants

Participants for this study included health providers (nurses and clinical officers) involved in the PIC$_4$C program at AMPATH and working in rural clinics and communities. These providers offered hypertension care to patients and were already using Samsung Galaxy Tab A7 tablets (Samsung Electronics Co., Ltd.) running the Android operating system, and which has the *mUzima* application installed. The providers used *mUzima* primarily for: (a) data entry and validation, (b) to retrieve and display historical data from the electronic AMPATH Medical Record System (AMRS, which is an instance of OpenMRS) [24], and (c) for decision support that provided prompts and reminders to guide the providers through the hypertension management algorithms [25]. The providers were expected to use *mUzima* as a point-of-care application during the patient encounter.

This study targeted to recruit all health providers who had been assigned mobile devices from which they used *mUzima* for patient data in the PIC$_4$C program. A total 23 providers were recruited representing 89 facilities and 24 community-based groups in two Kenyan counties. It was common for a provider to serve multiple facilities and communities.

## Study approval and informed consent

This study was approved by the Institutional Review and Ethics Committee (IREC) at Moi University School of Medicine in Eldoret, Kenya. To avoid possible change in behavior by providers when they are aware of being observed (Hawthorne effect), [26] the study was granted a partial waiver by the IREC. This waiver allowed the study to recruit and consent participants

to using mUzima, with a commitment and obligation of making participants aware of recording of log data at the end of the data collection period. Prior to participation in the study, written informed consent (IC) was obtained. Each consenting participant signed two copies of the IC form. One copy of the signed form was given to the study participant while the second was stored by the study team. All data collected were used exclusively for the study, and only availed to approved study team members.

## Study procedures

A trained research assistant recruited participants into the study, after which mobile devices used by the participants were upgraded with the new version of the mobile application that incorporated the logging capability. Usage logs were collected for a period of three months between Dec 2nd, 2019 and Mar 2nd, 2020, with collected log data stored in the secure *Event Log Server*. At the end of the study period, data were extracted from the collected usage logs and the AMRS EMR system. At the time of the study, no clear workday schedules were available for the clinicians, and individuals took two weeks off around the Christmas and New Year's holidays.

Log data extracted were filtered to exclude any test users. Needed data cleaning of the logs was done prior to analyses and included: (a) removal of any duplicate log records; (b) exclusion of logs that were not relevant to participants' app usage; (c) imputation of the correct timestamps (using operator and geolocation timestamps) for records from devices that had the wrong time set on the device; and (d) exclusion of records that were outside the study period. After data cleaning, the log data were transformed into patient-provider encounter records for analyses. Data from the EMRs included count and dates of relevant clinical encounter forms submitted by the study participants from the *mUzima* application.

## Data cleaning and analyses

Data cleaning and analyses were done using Python and Jupyter notebook [27–29]. Scripts were developed using *python* programming language to extract data from the MongoDB database, to clean and transform the data into encounter records, to generate datasets for the various metrics, to plot graphs for visualizations and to export the datasets into CSV file format for analyses (see **S3 Appendix** and **S4 Appendix** for sample scripts used). Python's Pandas library was used for manipulation of the data to clean it and generate the various datasets, while Matplotlib library was used for visualization.

The objective of this study was to leverage mobile app log data to better inform work patterns for care providers who do not work under the direct presence of their supervisors. Thus, metrics of interest that were derived from the collected data included: (a) number of patients seen per day—this was the total count of patients registered or whose encounter data were recorded in *mUzima* on a daily basis by the study providers; (b) number of days worked by providers during the study period–with dates where no trace of at least one record of patient encounter used as an indicator of no work performed on that day. Elaboration of days worked was disaggregated per study participant and by day of week; (c) work hours per day–represented by duration of time participants actively used the *mUzima* mobile application to record patient clinical data with primary goal of capturing work-related activities conducted outside of official work hours; and (d) length of patient encounters–representing the start to end recording of each patient encounter within *mUzima*. Prior to analyses to derive indicators, we assessed the relative usefulness of the log data using Pearson correlation coefficient by comparing captured logs for number of completed clinical encounters with those stored in the EMRs, as these two should correlate well.

## Results

Over the study period of 2$^{nd}$ December 2019 to 2$^{nd}$ March 2020, relevant usage logs were available for 13 of the 23 (56.3%) study participants, meaning that only this subset of study participants used the mobile application during the study period. In total, 39,229 log records were collected for the 13 study participants, and these included 2,497 clinical encounters.

### Distribution of work activity by number of days worked

Comparison of the total numbers of the work days for each participant as derived from the logs as well as from records in the EMRs showed a strong positive correlation between the two data sources (r(11) = .92, p < .0005). Workdays using mobile logs also included days where clinicians used mUzima to fill an encounter form but did not complete the form–these incomplete forms were not transmitted to the EMRs until they were completed, hence work days were higher from log-based data when compared to EMRs data. A workday was defined as a unique date with evidence of logs related to a patient clinical encounter.

**Fig 4** below presents the distribution of the total number of days worked by each provider as derived from the log data. There was wide variability in the number of days worked by providers, with only two of the thirteen providers (*USER_01* and *USER_02*) working for over 50%

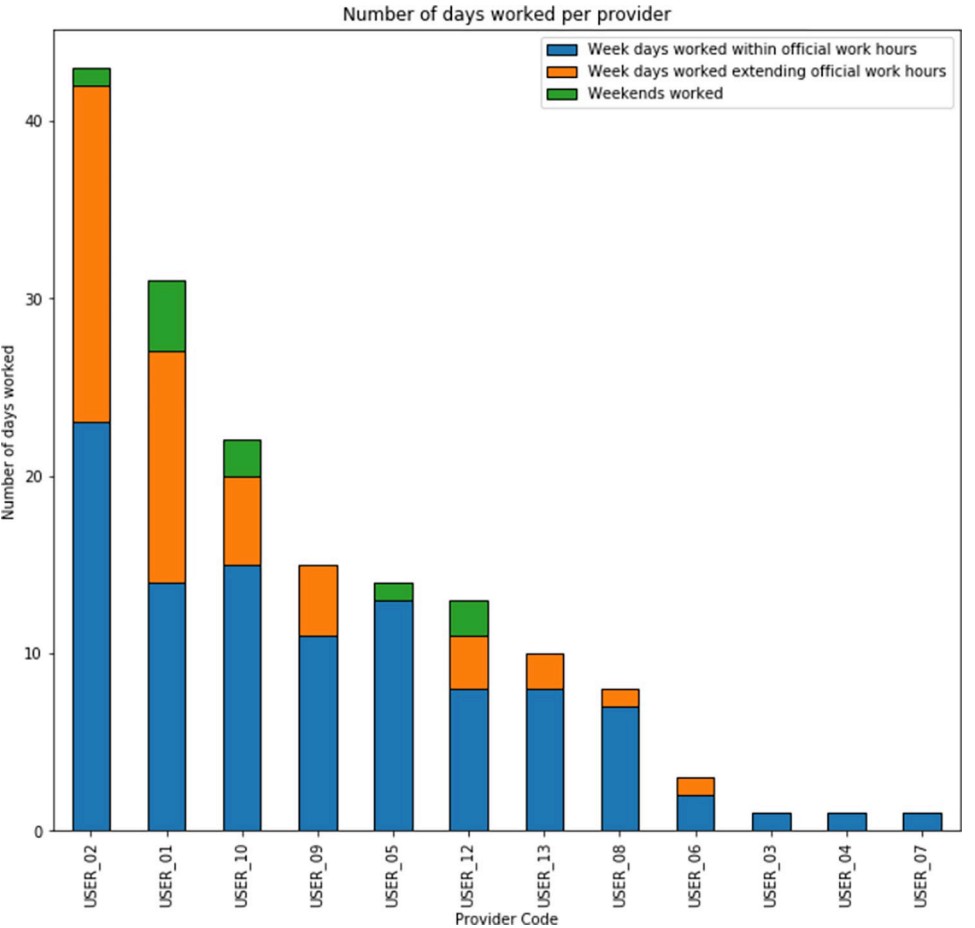

**Fig 4. Bar graph showing the number of days worked per study participant as per usage logs.**

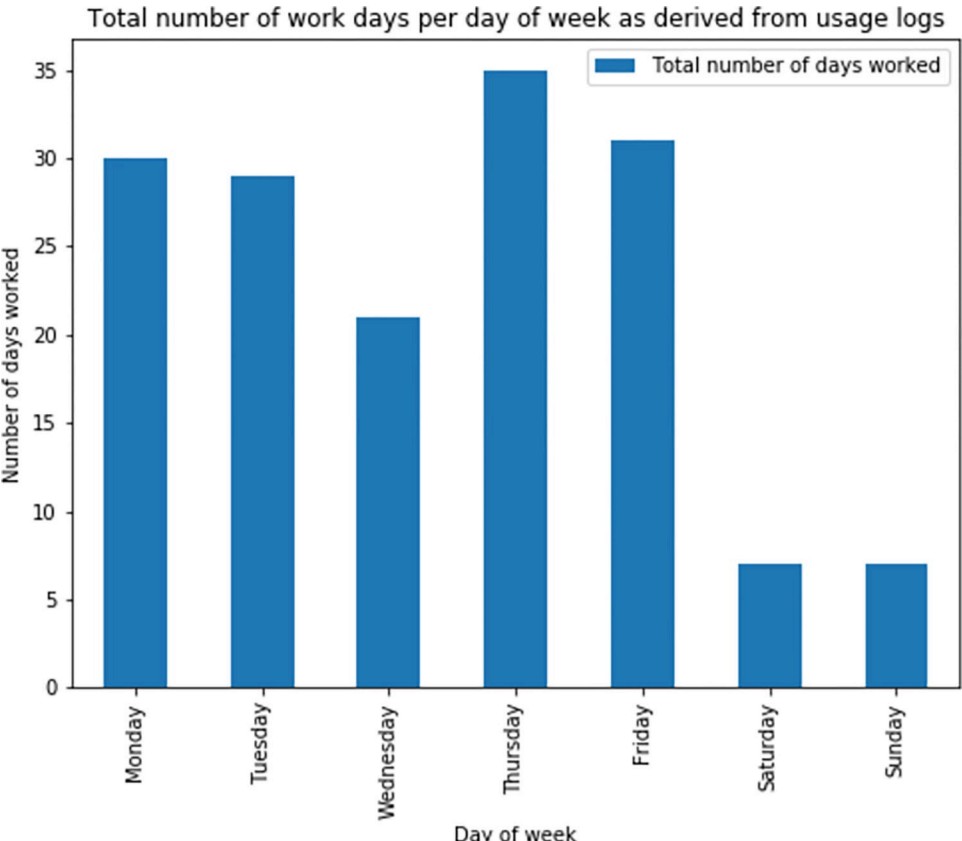

**Fig 5. Bar graph showing the total amount of workdays against day of week as derived from usage logs.**

of the expected workdays. Five providers (*USER_03*, *USER_04*, *USER_06*, *USER_07*, *USER_11*) had very few days recorded during the study period.

### Distribution of work activity by day of week

The work patterns by day of week were also analyzed (**Fig 5**). Breakdown of days worked by each participant did not show any clear patterns per worker of consistently missing work on particular workdays (e.g., Fridays). It was noted that five of the participants engaged in work during the weekends, contrary to the expectation of them being scheduled to work only from Monday to Friday (**Fig 4**).

### Average number of patients seen per day

The number of patients seen comprised the total count of patients registered or whose encounter records were logged on daily basis by each of the study participants. In total, 2,497 clinical encounters were registered during the study period, with 1,608 (64.4%) coming from just two users (*USER_01* and *USER_02*). On average, 14.5 (range 1–53) patients were seen per day by the providers (**Table 1**).

### Mobile application usage times and length of use

**Fig 6** provides a visualization of the work activities for all study participants by time of day, while **Fig 7** shows this visualization for one of the participants (USER_02). Through the logs, it

**Table 1. Total number of patients seen per study participant as derived from usage logs.**

| USER ID | Total number of patients seen |
|---------|-------------------------------|
| USER_01 | 371 |
| USER_01 | 1069 |
| USER_03 | 5 |
| USER_04 | 25 |
| USER_05 | 186 |
| USER_06 | 29 |
| USER_07 | 7 |
| USER_08 | 50 |
| USER_09 | 162 |
| USER_10 | 232 |
| USER_11 | 3 |
| USER_12 | 74 |
| USER_13 | 95 |

was evident that many of the providers continued to use the application into the late evening and early night, signifying that work was done by these providers beyond official work hours. A total of 563 (22.5%) of the encounter forms were entered after regular work hours, signifying that some of the providers used the application to enter data retrospectively- as opposed to using the application as a point-of-care system, where they would be able to take advantage of the provided decision support.

## Length to complete an encounter form

The duration of app usage per patient encounter session, and the time interval between app usage sessions were analyzed to determine the patterns of app usage as a surrogate for whether encounters were entered during actual encounter sessions or retrospectively. The results show

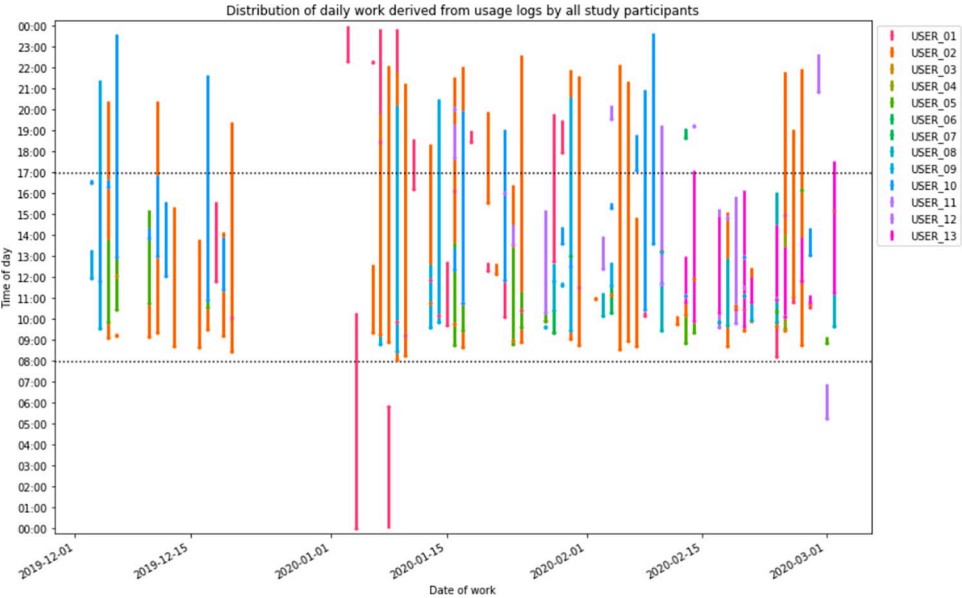

**Fig 6. Visualization of the distribution of workday activity by all study participants on days worked as derived from usage log data.**

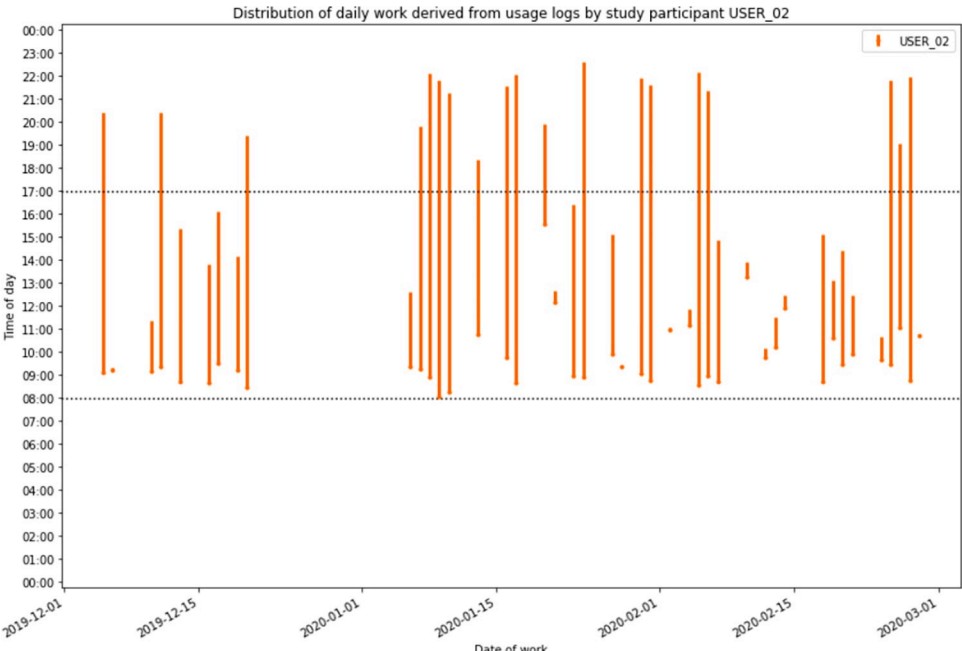

**Fig 7. Visualization of the distribution of workday activity by one study participant (USER_02) on days worked as derived from usage log data.**

that numerous encounter form entry sessions were conducted in a very short time duration, with short intervals between encounter form entry sessions (**Fig 8**). These short timeframes could only be achieved using retrospective data entry, rather that real-time entry during an actual patient encounter, as typical patient encounters take longer.

## Discussion

To our knowledge, this is the first study to leverage mHealth usage logs from healthcare providers in LMICs to inform work patterns and work performance by health workers. Comparison of mHealth-derived usage log metrics against EMR-derived metrics showed a strong positive correlation between number of days worked per participant. This supported reliability of leveraging usage logs for performance evaluation. mHealth-derived usage logs have particular relevance for metrics that are usually not collected or available in EMRs, such as length of patient encounter, workday length, and work hours by providers. Even for metrics such as number of patients seen, log data perform better as they also capture incomplete encounters.

The described work provides an additional approach that can be used to evaluate work performance. Historically, work performance evaluations for health providers have involved time-and-motion studies, which often require human observers and/or manual recording of activities by the providers being assessed [30–32]. Further, use of mHealth paradata for work performance assessment innovatively extends role of mHealth solutions to better support key functions of information technology in LMICs, which is to strengthen the health systems [13,33,34]. By re-using paradata collected as providers use the mHealth application, this approach promises to be cheaper and more scalable than traditional approaches that require human involvement to assess work patterns. However, use of paradata must take into strong consideration the ethical, legal and social implications (ELSI) of paradata use. Of particular importance is the need for informed consent for the providers, with mechanisms adopted to

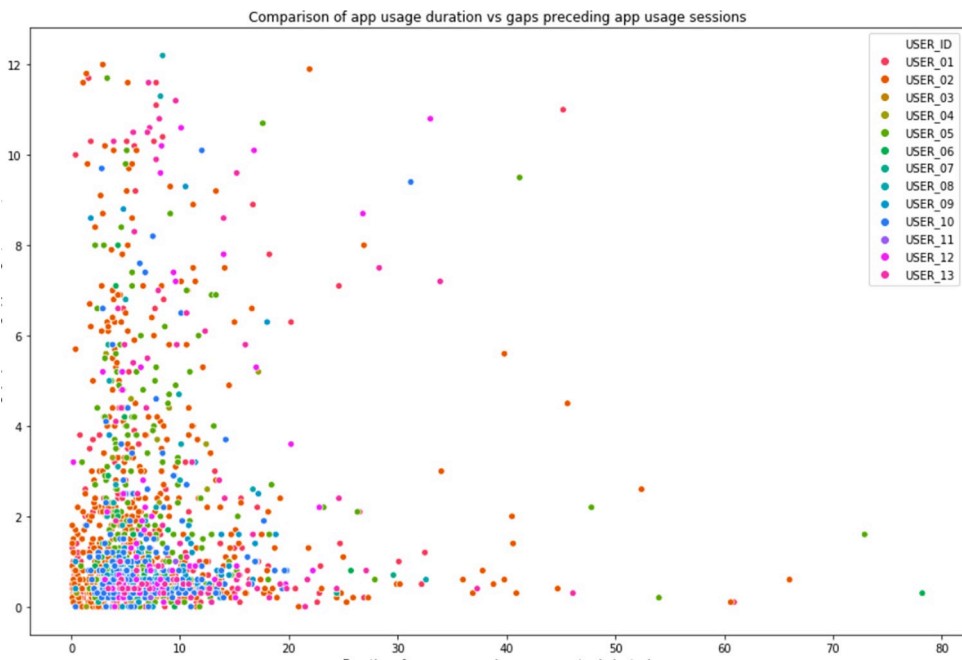

**Fig 8. Encounter entry duration and time between encounters.**

ensure that the paradata and knowledge derived from them are used strictly to inform and incentivize care providers and for supportive supervision, and never used for punitive purposes.

In the current study, we observed that healthcare providers often worked outside regular work hours and during weekends. Having this knowledge on hand equips care programs to investigate factors that contribute to these work patterns and seek remedies. It is possible that some health workers are simply overwhelmed during the day and resort to taking work home with them. Alternatively, some workers might be slow or uncomfortable with using mHealth technologies as a point-of-care system during patient visits. As such, the providers might prefer to retrospectively enter clinical data in the application–and this would lead to longer work hours, risking provider burnout. The finding that the mHealth application was not being used as expected (i.e., as a point-of-care system) has important implications—it means that the care providers are not taking advantage of real-time computerized clinical decision support features within the application that are relevant to improving quality of care [25]. These decision-support features have particular relevance in LMIC settings that employ task-shifting of care services to lower-cadre staff.

Differences between providers in number of days worked, average number of patients seen per day and hours worked offer insights on areas where individual health worker performance can be improved. In the age of COVID-19 pandemic, where in-person supervision can be particularly challenging in some settings, it is important to have mechanisms that provide insights into worker performance for timely supportive interventions. Further, the derived paradata-derived work performance metrics (e.g., patients seen, days worked and work day length) can be innovatively leveraged to improve performance, motivation and self-efficacy of health workers working in remote facilities through approaches such as gamification and ecological momentary interventions [35–37].

Several limitations in our study deserve mention. The generalizability of our findings is limited by the fact that it involved one mHealth application, a few facilities in a single country,

and with limited number of healthcare providers. Our evaluation also only lasted for a short period of time, and we cannot account for possible changes in work performance occasioned by other factors. However, the study achieved its main goal of evaluating feasibility of using mHealth-derived paradata to evaluate health worker performance and provides a re-usable approach for other mHealth applications and settings. Results of this study have been shared with the clinical team to help inform work patterns and approaches for improving them. As the next steps, we will employ qualitative approaches to better understand reasons for the observed work patterns and for the variations in performance between providers. We will also leverage the paradata-derived metrics to provide real-time visualization to providers for insights on their own individual performance, and their performance relative to their colleagues. Finally, we plan to incorporate timely automated feedback and support mechanisms based on derived performance metrics to supplement support where direct supervision is a challenge.

## Conclusions

mHealth paradata can be used to derive work-performance metrics for providers working in disconnected LMIC settings. This approach extends the use of mHealth applications in the area of health systems strengthening and is easily scalable for supportive supervision.

## Supporting information

**S1 Appendix. Description of fields comprised in a usage log.**
(DOCX)

**S2 Appendix. Types of usage logs recorded.**
(DOCX)

**S3 Appendix. Excerpts of python scripts used for data extraction.**
(DOCX)

**S4 Appendix. Excerpts of python scripts used for data cleaning.**
(DOCX)

**S5 Appendix. Excerpts of python scripts used for generating encounter records.**
(DOCX)

## Acknowledgments

The authors thank the participating healthcare providers and clinics, as well as the Primary-health Integrated Care project for Chronic diseases (PIC$_4$C) project at AMPATH, Kenya.

The contents are solely the responsibility of the authors and do not necessarily represent the official views of USAID, Norad, or the United States Government.

## Author Contributions

**Conceptualization:** Simon Savai, Jemimah Kamano, Lawrence Misoi, Peter Wakholi, Md Kamrul Hasan, Martin C. Were.

**Data curation:** Simon Savai, Lawrence Misoi, Md Kamrul Hasan.

**Formal analysis:** Simon Savai, Md Kamrul Hasan.

**Funding acquisition:** Martin C. Were.

**Investigation:** Simon Savai, Jemimah Kamano, Lawrence Misoi, Peter Wakholi, Md Kamrul Hasan, Martin C. Were.

**Methodology:** Simon Savai, Peter Wakholi, Md Kamrul Hasan, Martin C. Were.

**Project administration:** Simon Savai, Jemimah Kamano, Lawrence Misoi, Martin C. Were.

**Supervision:** Jemimah Kamano, Peter Wakholi, Martin C. Were.

**Validation:** Jemimah Kamano, Lawrence Misoi, Md Kamrul Hasan.

**Visualization:** Md Kamrul Hasan.

**Writing – original draft:** Simon Savai, Martin C. Were.

**Writing – review & editing:** Simon Savai, Jemimah Kamano, Lawrence Misoi, Peter Wakholi, Md Kamrul Hasan, Martin C. Were.

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
