## [Decision Letter · Decision Letter 0]

22 Jun 2022

PDIG-D-22-00108

Leveraging mHealth Usage Logs to Inform Health Worker Performance in a Resource-Limited Setting: Case Example of mUzima use for a Chronic Disease Program in Western Kenya

PLOS Digital Health

Dear Dr. Were,

Thank you for submitting your manuscript to PLOS Digital Health. After careful consideration, we feel that it has merit but does not fully meet PLOS Digital Health's publication criteria as it currently stands. Therefore, we invite you to submit a revised version of the manuscript that addresses the points raised during the review process.

Please submit your revised manuscript within 30 days . If you will need more time than this to complete your revisions, please reply to this message or contact the journal office at digitalhealth@plos.org. Please include the following items when submitting your revised manuscript:

We look forward to receiving your revised manuscript.

Kind regards,

J Mark Ansermino, MBBCh

Section Editor

PLOS Digital Health

Journal Requirements:

Please state what role the funders took in the study. If the funders had no role in your study, please state: “The funders had no role in study design, data collection and analysis, decision to publish, or preparation of the manuscript.”

2. Please send a completed 'Competing Interests' statement, including any COIs declared by your co-authors. If you have no competing interests to declare, please state "The authors have declared that no competing interests exist". Otherwise please declare all competing interests beginning with the statement "I have read the journal's policy and the authors of this manuscript have the following competing interests:"

3. In the online submission form, you indicated that "Data will be availed with appropriate approvals.". All PLOS journals now require all data underlying the findings described in their manuscript to be freely available to other researchers, either 1. In a public repository, 2. Within the manuscript itself, or 3. Uploaded as supplementary information.

4. Please provide separate figure files in .tif or .eps format and remove any figures embedded in your manuscript file. Please also ensure that all files are under our size limit of 10MB.

For more information about how to convert your figure files please see our guidelines: https://journals.plos.org/digitalhealth/s/figures

5. We do not publish any copyright or trademark symbols that usually accompany proprietary names, eg (R), (C), or TM (e.g. next to drug or reagent names). Please remove all instances of trademark/copyright symbols throughout the text, including © World Bank on page 25.

Additional Editor Comments (if provided):

Reviewers' comments:

Reviewer's Responses to Questions

**Comments to the Author**

1. Does this manuscript meet PLOS Digital Health’s publication criteria? Is the manuscript technically sound, and do the data support the conclusions? The manuscript must describe methodologically and ethically rigorous research with conclusions that are appropriately drawn based on the data presented.

Reviewer #1: No

Reviewer #2: Yes

2. Has the statistical analysis been performed appropriately and rigorously?

Reviewer #1: Yes

Reviewer #2: Yes

3. Have the authors made all data underlying the findings in their manuscript fully available (please refer to the Data Availability Statement at the start of the manuscript PDF file)?

Reviewer #1: No

Reviewer #2: Yes

4. Is the manuscript presented in an intelligible fashion and written in standard English?

Reviewer #1: Yes

Reviewer #2: Yes

5. Review Comments to the Author

Reviewer #1: The authors report the results of a study conducted at a chronic disease program in Kenya. The study participants used an mHealth application (mUzima) during clinical care. In addition to having a decision support system, the app also captured usage logs. The objective of this study was to valuate usefulness of mHealth usage logs to inform health worker performance.

I find the paper interesting and useful. However, I believe the paper is incomplete. Although the authors’ stated objective is to valuate usefulness of mHealth usage logs to inform health worker performance, they did not achieve that. After presenting the data analysis, the authors leave it to future work to reasons of the findings and how the findings could be used to improve the health delivery process. (they state in lines 345-351 that “As the next steps, we will employ qualitative approaches to better understand reasons for the observed work patterns and for the variations in performance between providers. We will also leverage the paradata-derived metrics to provide real-time visualization to providers for insights on their own individual performance, and their performance.”) Please note that the study was completed more than two years ago. Although the results are generalizable, results would be valuable as a case study if they could explain how and why they did or did not use the findings to improve the care delivery process.

In addition to this, it would be useful if they provide the following information: 

• What are the work requirements for each worker? Do the workers use the app only outside of their facilities? It is mentioned (line 229) that in total, 39,229 log records were collected for the 13 study participants, and these included 2,497 clinical encounters. It does not mention how many workdays are included in this three-months study period. My rough calculation leads to 115 workdays per worker giving a total of about 1485 workdays. However, from figure 5, total number of resulting workdays is only about 160 workdays, which is only 10% of the workdays. It is important that the authors justify this.

• Based on figure 3, newly collected patient data from the app is automatically uploaded to EMR. If so, the correlation coefficient should be 1, not 0.92. 

• Figure 6 supersedes figure 4 and hence delete Figures 4. Also, it is not necessary to present user numbers sequentially. A Pareto chart is much more meaningful here.

• Figures 8 and 9 are not informative as presented. If the objective is to visualize the variation in length of workdays, a simple relative frequency distribution of length of workdays would be useful. Note that the actual dates are unimportant. You can also have two relative frequency distributions in figures 8 and 9 in one figure.

Reviewer #2: The authors present a novel use of usage log data to track health provider activity (a surrogate for performance) while using a mobile application (mUzima) that is linked to a larger electronic medical record system within the primary health care system in several counties in Western Kenya. The manuscript is well written, and the authors have provided adequate information regarding their methods and analyses. The additional appendices were also helpful to understand the scope of the data they gathered from the system for their research.

The authors conclude that they were able to reliably demonstrate that the log data can provide detailed information of the days and hours worked, number of patients seen per day, and how the application was being used to inform many areas of improvement related to health worker performance. They also observed large differences in work performance between health providers, and work performed outside of official work hours and on weekends and the authors also determined that many (if not all) of the health providers also used the application sub-optimally for entering patient data after the patient visit, as opposed to using the application during the patient encounter to best leverage built-in clinical decision support functionality. 

The research would have been strengthened by using a mixed-methods approach to also interview the providers who participated in the study to better understand what factors impacted their work performance (barriers such as workload, lack of access to the mobile network/mobile data credit to utilize the application in real time, and/or functionality of the mUzima application itself etc), however, the COVID-19 pandemic may have made this complementary data collection possible (physical distancing etc) as the study data was collected in the 3 months leading up to the declaration of the global pandemic (March 2020). The authors state in their discussion that they recognize that the lack of qualitative data is a weakness/limitation and that they plan to explore qualitative approaches to better understand their research findings.

The authors used the usage log data as a surrogate for ‘work performance’. While this is a reasonable surrogate to track clinical activities (as mUzima links to the EMR) and using the usage log data can serve as a surrogate for work performance, they are not entirely equal. The authors have addressed this adequately within their discussion and they have also highlighted the ethical implications of using these types of approaches strictly to inform, support, and incentivize care providers and not use the data for punitive purposes.

Overall, the research is novel and serves as an excellent jumping off point for further research in this area. It has significant potential to be utilized and leveraged further for future efforts and strategies to improve health worker performance and strengthen the use of mHealth approaches for health system strengthening, especially within chronic disease management in primary care.

I did note one error/oversight in the reference list. Reference #13 and #35 are the same reference. This needs to be corrected.

6. PLOS authors have the option to publish the peer review history of their article (what does this mean?). If published, this will include your full peer review and any attached files.

**Do you want your identity to be public for this peer review?** For information about this choice, including consent withdrawal, please see our Privacy Policy.

Reviewer #1: No

Reviewer #2: No

---

## [Editor Report · Decision Letter 1]

25 Jul 2022

Leveraging mHealth Usage Logs to Inform Health Worker Performance in a Resource-Limited Setting: Case Example of mUzima use for a Chronic Disease Program in Western Kenya

PDIG-D-22-00108R1

Dear Dr Were,

We are pleased to inform you that your manuscript 'Leveraging mHealth Usage Logs to Inform Health Worker Performance in a Resource-Limited Setting: Case Example of mUzima use for a Chronic Disease Program in Western Kenya' has been provisionally accepted for publication in PLOS Digital Health.

Best regards,

J Mark Ansermino, MBBCh

Section Editor

PLOS Digital Health

Thank you for your revision. We look forward to more detail on the qualitative component!